# Numerical Modeling of Masonry Infilled Reinforced Concrete Building during Construction Stages Using ABAQUS Software

**Boudjamaa Roudane \***, **Süleyman Adanur and Ahmet Can Altunışık**

Department of Civil Engineering, Karadeniz Technical University, 61080 Trabzon, Turkey
* Correspondence: rou.ingc@ktu.edu.tr

**Abstract:** The effects of seismic actions on reinforced concrete (RC) structures are strongly influenced by the dynamic behavior of their materials. It is crucial to find a simple definition of the natural frequencies of reinforced concrete buildings, particularly in relation to both principal and secondary elements constructing the reinforced concrete building type. This paper firstly presents a comparison with the ambient vibration surveys. An analysis model of different stages of construction of the reinforced concrete masonry wall was compared using the finite element software. In the second step, structural responses of the model were investigated by means of static analysis. Three main types were examined: Bare frame for one, two and three storeys; brick-walled; and coated cases. Modal analysis is carried out by ABAQUS software starting from the deformed building, to provide the natural frequencies and mode shapes. For the natural frequencies, a good agreement is obtained between analytical and experimental results. Furthermore, the comparison results between different cases show that the application of the plaster work increases the lateral stiffness and has significant effects on the dynamic response of the buildings.

**Keywords:** ABAQUS software; construction stages; dynamic characteristics; reinforced concrete building; plaster work

---

## 1. Introduction

Assuring the highest level of safety is significant from an economic and strategic view. This applies to both historical and new reinforced structures against dynamic loading, for example, earthquakes. This can be accomplished if the dynamic characteristics of the buildings such as natural frequencies and mode shapes are firmly determined. Modal analysis is a technique that evaluates the modal parameters of a structure. These parameters are key to the determination of the finite element model updating, structural control, damage detection, and long-term structural health monitoring.

The evaluation of the dynamic characteristics using ambient vibration measurements is a tough task due to the presence of noise having dominant frequencies that could be involved in the modes of vibration, especially when the position and orientation of the sensors are not well placed. This noise effect could be resolved by conducting various tests and putting the sensors in the optimum setting. After that, the results obtained from the measurements can be considered almost accurate [1]. The application of well-known different identification techniques include: The Crystal-Clear Stochastic Subspace Identification (CC-SSI) method [2], the Stochastic Subspace Identification (SSI) method [3], and the Enhanced Frequency Domain Decomposition (EFDD) method [4]. The use of the ambient vibration measurements has been proven to be relatively fast, cost reducing and economically efficient for the determination of dynamic characteristics [5].

The experimental measurements taken during different stages of construction provide more information for the structure compared to the ones obtained from only a single measurement at the completion stage of building. There have been a few similar studies conducted in the past [6].

Rombach [7] concluded that one of the most important advantages of construction-stage analysis is that the engineer can examine stress distributions as well as deformations at different levels of the construction sequence. Arslan and Duemuş [8] and Kaplan et al. [9] determined natural frequencies, mode shapes and damping ratios of full-scale one-storey one-bay infilled RC frames for bare frame, brick infilled and brick infilled with plaster using the Operational Modal Analyses method under ambient vibration. Results demonstrated that the dynamic characteristics change fundamentally relying upon the existence of an infill wall and plaster, and should be considered in structural analysis. Experimental and analytical approaches have been noted in many masonry-infilled RC buildings researchers such as the evaluation of dynamic characteristics (natural frequency in particular). Bayraktar et al. [10] found that the first measured natural frequencies of three buildings which consisted of three construction stages named bare frames, infill walls, and completed building stages, were greater than the calculated frequencies. Al-Nimry et al. [11] investigated the impact of cracking of the six-storey building in the period of vibration. Bikçe et al. [12] showed that the period obtained from the analytical results increased half a percent with both the measured and code values. Amanat and Hoque [13] used modal analysis to determine the fundamental frequency of vibration of several forms of RC-framed buildings infills and concluded that the primary parameters affecting the period are the height, the number, length of bays, and the amount of infills. Also, they observed that the infills' presence increased the frequencies by about 30%. Practically, most of the earthquake regulations neglect the effects of the infilled plaster work and design process [14].

The finite element method is a useful tool for engineers to assess, clarify and comprehend the conduct of any structural system. Finite element method (FEM) is a numerical solution method which looks for an approximate solution for various engineering problems. Timurağaoğlu et al. [15] studied the behavior of reinforced concrete frames with brick and gas concrete infill walls using analytical and finite element methods. The one-storey reinforced concrete frame system, which has been experimentally studied before, was modeled with the help of the computer program. The results obtained from the analysis were compared with the experimental ones and the effects of the infill wall to the frame behavior were examined. In addition, the different equivalent compression bar models available in the literature were modeled in the computer environment with the help of finite element method, and the analysis results were compared with those obtained from experiments. Kubalski et al. [16] adopted numerical models in order to describe the inelastic behavior of the system, as demonstrated by the acquired results of the overall structural response as well as the damage propagation within the infill wall. Comparing between experimental and numerical techniques underlined the valuable contribution of validated numerical simulations to infer rational recommendations for the design of masonry infilled frames. Khatiwada and Jiang [17] demonstrated in the study the development of a finite element model of the infilled frame with the masonry wall under monotonic lateral loading. The force-displacement curve, failure mode and crack pattern demonstrate that the model is able to simulate the behavior of an infilled frame. The study also shows that the simplified micro modeling technique with cohesive surface is relevant in reproducing extensive outcomes. Furtado et al. [18] studied the impact of the infill masonry walls' presence in the seismic performance of a 15-storey high-rise building situated in Nepal. They utilized the obtained results to calibrate the numerical model built in the software SAP2000 [19]. The researchers carried out linear elastic analyses to evaluate the effect of the infill panels in the expected dynamic response of the structure. The panels' presence increased the storey shear and the maximum base shear by about 20%. It also increased the torsion effect. In recent years, the effects of plaster work on the seismic behavior of infilled frames have been investigated by several researchers. Plastering significantly affects the behavior of infill walls and also increases the natural frequency, stiffness and strength of the structures [20–22].

The numerical modeling necessary to determine the building's dynamic properties is similar to the model used to predict 'static' behavior or to evaluate safety under given loads; therefore, the results go beyond assessing dynamic behavior. This study aims to:

1.  Provide information about the dynamic characteristics of the Turkish RC masonry buildings at different construction stages.
2.  Determine the modeling errors through comparison study results found in the laboratory.
3.  Understand the effect of the structural component and material such as wall and plaster work on dynamic characteristics of the completed building.
4.  Compare the ultimate displacements and principal stresses for each stage.

In this study, numerical modeling has been conducted by using ABAQUS software at each stage. The finite element model is described, and the main calculated results are presented. The measurements and calculations are then compared to provide a better understanding of the building behavior and to identify where the numerical model needs to be improved.

*Novelty of This Study Paper*

The main contribution of this paper is to investigate both analyses (modal and static) of the reinforced concrete (RC) frame building during different construction stages, taking into consideration the effect on the modeling masonry infill (MI) and plastering brick walls, as well as a comparison between experimental and analytical results of the fundamental frequencies is performed. An analytical model analysis is performed on the three-dimensional finite element model developed using the ABAQUS software package.

**2. Description of the Structure**

As a case study, an infilled reinforced building masonry prototype was built. The building was tested through Operational Modal Analysis (OMA). In the work experimental research, dynamic characteristics of a three storey reinforced concrete structure with a 1/2 scale during different contruction stages were determined [23]. Enhanced Frequency Domain Decomposition (EFDD) technique was used to obtain frequencies and mode shapes.

*2.1. Geometry of the Prototype*

The prototype is a 1:2 scaled three-storey building, as can be seen in Figure 1, aimed at reproducing the typical RC masonry buildings of Turkey. The construction stage analysis was constructed by adding one storey to the structure at each stage. In the third storey, for the whole building, the brick walls were built, and at the last stage, all exterior walls were coated by the cement plaster. All stages were done on one completed model. The typical storey framing consists of reinforced concrete slabs, and sections of $150 \times 200$ mm for columns and beams, which are shown in Figure 2b, with constant size at all storeys. The thickness of the brick walls and the cement plaster are 190 mm and 20 mm, respectively. The main construction stages of the buildings can be classified into three stages: Bare frame, brick-walled and coated brick with dimensions of 2.5 m length, 1.5 m width and 4.8 m height. Other dimensions and section properties of the RC building elements are given in Table 1.

**Table 1.** Dimensions and section properties assigned to the RC building elements.

| Column and Beam (mm) | | Foundation (mm) | Slabs (mm) |
| --- | --- | --- | --- |
| Reinforcements | Dimensions | Thickness | |
| 4 Φ 10 | $150 \times 200$ | 500 | 75 |

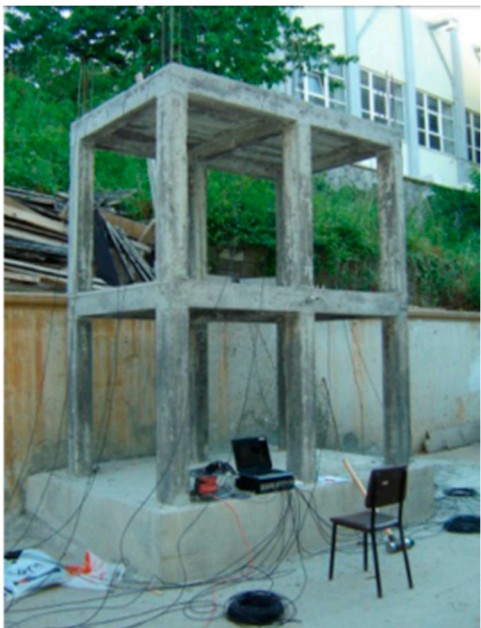

**Figure 1.** Test set-up for the two-storey building model.

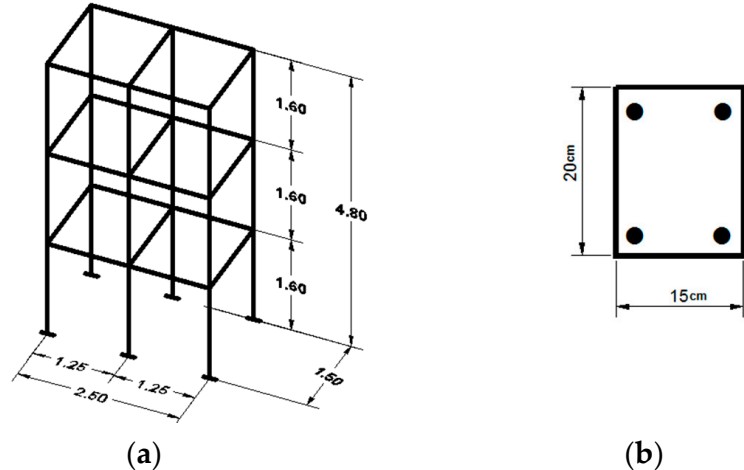

| (a) | (b) |
|:---:|:---:|

**Figure 2.** The dimensions and sections of the RC building model. (**a**) 3D-view of three storeys; (**b**) the beam and column sections.

*2.2. Material Properties*

The compressive strength of masonry walls is taken as 5 MPa, considering typical solid bricks and mortar used in Turkey. Furthermore, according to Turkish Standard Institution (TS-705) [24], vertical perforated bricks have a density between 1800 and 2200 kg/m$^3$. Material properties used in the initial finite element model of the mortar of the plastering were taken from earlier studies [25].

Table 2 summarizes the basic material properties used in the building. The properties of the different materials of RC building models, namely elasticity modulus, Poisson's ratio, and density, which will be taken into consideration in the analysis, are crucial for an accurate FE model. The elastic modulus of masonry units was calculated using the formula below [26].

$$E_d = 200f_d \tag{1}$$

where $E_d$ is the elasticity modulus, and $f_d$ is the compressive strength of masonry units.

**Table 2.** Material properties used in the initial finite element model.

| Elements | Modulus of Elasticity (MPa) | Poisson's Ratio (-) | Density (kg/m$^3$) | Compressive Strength (MPa) |
|---|---|---|---|---|
| Concrete | 22,000 | 0.20 | 2350 | 20 |
| Steel | 210,000 | 0.30 | 7850 | 420 |
| Brick | 1000 | 0.20 | 2000 | 05 |
| Cement plaster | 10,000 | 0.20 | 1950 | 7.50 |

### 2.3. Finite Element Modeling

The purpose of the analytical studies is to validate the operational modal analysis carried out in Türker and Bayraktar [23]. Analytical studies were conducted through the linear models using the finite element method which uses the ABAQUS software [27], in order to evaluate the suitability of the dynamic behavior of the building. For numerical modeling, it is critical to comprehend which components and materials of the structure have structural importance.

In this paper, depending on the physical and mechanical properties of the building, the program is utilized to decide the natural frequencies and the corresponding mode shapes of the different types of the buildings for construction-stage studies. In order to acquire precise outcomes from the FE model, all components in the model were deliberately relegated to the same mesh size, which ensures every two distinct materials share the same node. The ideal component sizes were picked as roughly 75 mm. The mesh elements utilized for modeling concrete and reinforcement bars are eight-node solid brick element (C3D8R) and the two-node linear beam elements (B31) as demonstrated in Figure 3, respectively. In the brick-walled case, the interaction between the frame and the infill plays a crucial role in the behavior of infilled frames. Ordinarily, the brick walls are connected to RC frames using mortar joints, by these articulations, the interaction between the wall and the frame is fulfilled. The general contact between surfaces was named "surface to surface", In this contact, the slave surface is chosen as masonry wall with fine-mesh while master surface is chosen as RC frame.A macro modeling technique was utilized for the masonry walls, which results in less time and computational effort [28–30].

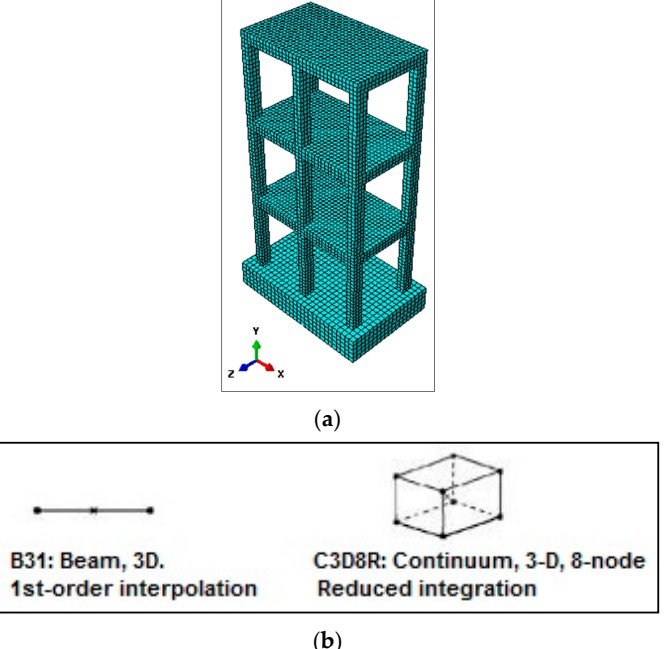

(**a**)

(**b**)

**Figure 3.** (**a**) FE model of building. (**b**) Solid finite element model Shows the element adopted by ABAQUS.

In this technique, there is no dissimilarity between mortar joints and brick units. Also, a homogeneous isotropic material constitutes the masonry assemblage. The three-dimensional finite element modeling was implemented using the ABAQUS/Standard to simulate the dynamic behavior of the investigated building. To set up this functionality, the frame and the wall are connected by interface/constraint elements that can transfer normal and shear stresses. FE modeling of the frame wall interface is capital due to its impact on the whole reaction of the frames under seismic loading. Figure 2 demonstrates the typical mesh of the FE model for the whole building. Modeling and mesh generation are established using the same methods for all model types. In order to assume an ideal bond between the reinforcement and concrete, an embedded connection was used. 3-D truss elements representing the reinforcement are the embedded region, while the concrete (foundation, beams and slabs) is the host region. Boundary conditions for the model are assumed to be fixed to the ground, and finally the model is delivered to the solver.

In the conventional structural design method, soil–structure interaction (SSI) effects are not considered [31]. Neglecting the (SSI) effect for a relatively flexible structure founded on hard soil is reasonable. However, for a relatively stiff structure founded on either soft or medium soil, neglecting SSI has a great impact on structural response and design [32]. The model was built in a laboratory and is based on a rocky ground. The soil can be considered as hard. Therefore, in this study, the soil factor is not taken into consideration.

## 3. Results and Discussion

Five main construction stages were considered by different structural stages. Firstly, modal analysis was determined. Secondly, static analysis was carried out.

### 3.1. Modal Analyses

The modal analysis results will be presented and discussed throughout the present section. The comparison and discussion of the modal analysis results from the five numerical models will be presented in terms of natural frequencies. Thus, corresponding mode shapes were identified.

### 3.1.1. The Natural Frequencies

A summary of the relative errors between experimental and numerical frequencies of all cases: bare frame (one-storey, two-storey and three-storey cases), brick walled and coated case are presented in Table 3.

It is shown that the first three analytical frequencies of frame bare cases, mainly (two- and three-storey cases) have considerably similar tendencies; within relative errors less than 6% compared to field measurement results. From the comparison between (three-storey and brick-walled cases) frequencies, it can be observed that, as expected, the frequencies of the model without infill masonry walls (three-storey case) are quite a bit lower than the experimental ones. The first and second frequencies are 70% lower. This is justified by the absence of infill masonry walls which resulted in the reduction of the global structure lateral stiffness. Regarding the model considering the coated wall case, it is observed that the numerical frequencies are higher than the experimental ones. There were some errors and a perfect match was not obtained. This could be due to the uncertainty of mechanical parameters of cement plaster work, so a subsequent FE model updating analysis is necessary for the coated-brick wall case (the relative error > 5%), in order to minimize the differences between the analytically- and experimentally-obtained dynamic properties by uncertain modeling parameters such as material characteristics of cement plaster and the effect of the soil-building interaction.

**Table 3.** Relative errors (%) between experimental and numerical frequencies.

| Construction Stage Cases | Vibration Modes | Frequency (Hz) | | | | Relative of Errors (%) |
|---|---|---|---|---|---|---|
| | | Experimental | | Analytical | | |
| One-storey case | 1 | 18.91 | | 18.83 | | 0.42 |
| | 2 | 23.40 | | 22.63 | | 3.40 |
| | 3 | 28.72 | | 28.37 | | 1.23 |
| Two-storey case | 1 | 9.69 | | 10.18 | | 4.81 |
| | 2 | 11.36 | | 11.54 | | 1.56 |
| | 3 | 14.08 | | 14.79 | | 4.80 |
| | 4 | 27.33 | | 28.76 | | 4.95 |
| | 5 | 34.51 | | 36.35 | | 5.06 |
| Three-storey Case | 1 | 6.43 | | 6.98 | | 3.58 |
| | 2 | 7.32 | | 7.47 | | 2.00 |
| | 3 | 9.33 | | 9.84 | | 5.18 |
| | 4 | 19.70 | | 20.48 | | 3.81 |
| | 5 | 23.49 | | 23.09 | | 1.73 |
| Brick-walled case | 1 | 11.18 | | 11.96 | | 0.58 |
| | 2 | 13.17 | | 12.97 | | 1.54 |
| | 3 | 20.67 | | 20.05 | | 3.09 |
| | 4 | 37.76 | | 36.47 | | 3.53 |
| | 5 | 44.35 | | 42.90 | | 5.80 |
| Coated-wall Case | 1 | 13.89 | | 14.54 | | 4.47 |
| | 2 | 16.79 | | 17.25 | | 2.67 |
| | 3 | 31.23 | | 29.68 | | 5.22 |
| | 4 | 52.01 | | 44.62 | | 16.56 |
| | 5 | 56.80 | | 50.67 | | 12.10 |

It can be seen in Figure 4 that the greater values of natural frequencies are most manifest in the coated-wall case. All exterior surfaces of the building were plastered with mortar. Because of the plaster layer, the frequencies increase about 21%. However, in the analyses, the effect of plastering into the inner and outer infill walls of RC buildings is commonly neglected by structural engineers, even though it plays a significant role in the dynamic behavior of the structure.

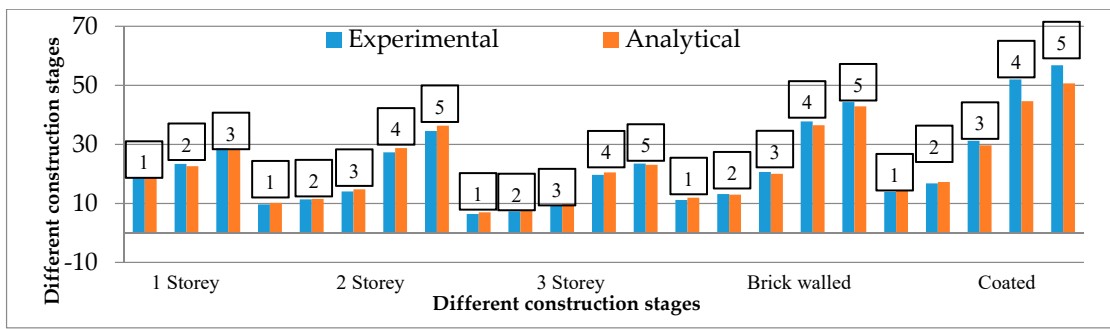

**Figure 4.** Comparison between natural frequencies for construction stages.

### 3.1.2. Mode Shapes

Due to reasons beyond our control (the absence of the complete measurement file taken from vibration measurements), the modal assurance criterion (MAC) analysis could not be performed. The results of the mode shape pattern up to the first three modes of the different structural stages: Frame-bare cases, brick-walled and coated-wall cases are given in Tables 4 and 5 respectively.

**Table 4.** First three mode shapes identified analytically of the one storey, two storey and three storey cases: (a) 1st Longitudinal mode, (b) 2nd Transverse mode, (c) 3rd Torsional mode, respectively.

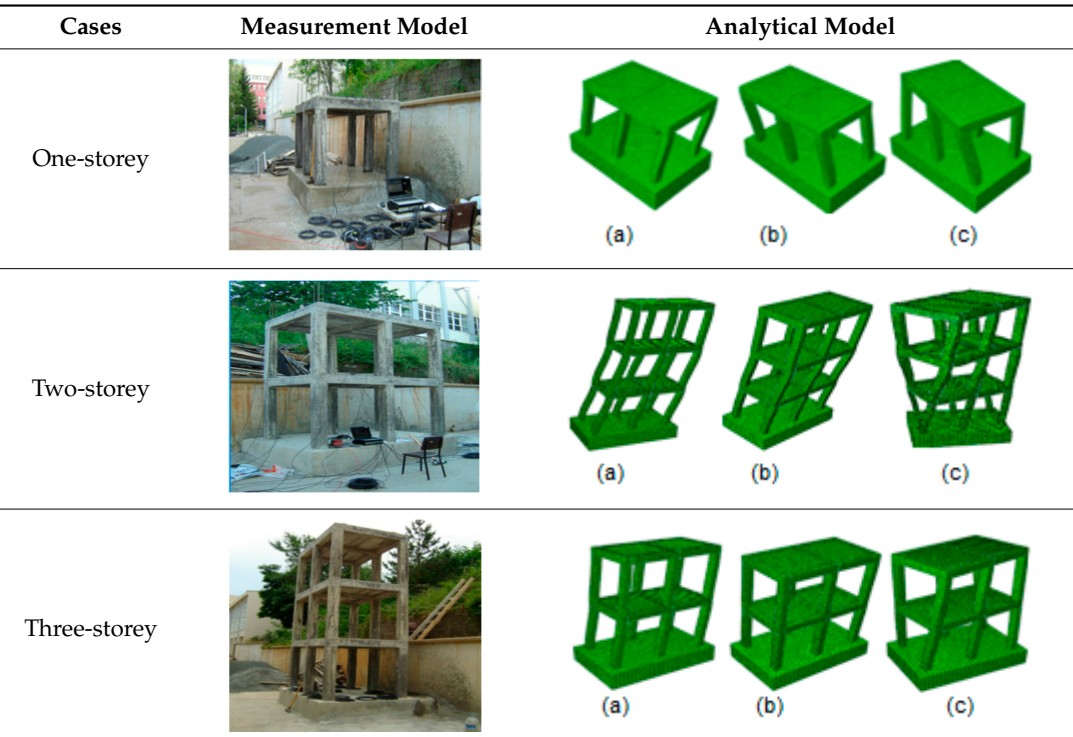

| Cases | Measurement Model | Analytical Model |
|---|---|---|
| One-storey | | (a) (b) (c) |
| Two-storey | | (a) (b) (c) |
| Three-storey | | (a) (b) (c) |

The modal behaviors of the first-, second- and third-storey cases (Table 4) are longitudinal, transverse and torsional. However, it can be seen that the mode shapes of the brick walled and coated-wall cases (Table 5) change significantly, as the transverse, longitudinal for first and second mode, respectively. The non-structural elements have contributed highly to the lateral stiffness, even changing the direction of the mode shape of a structure [33]. The locations of the infill in the two bays (traverse direction) mainly contribute to increase the global stiffness and strength of the structure in the transverse direction, compared to the longitudinal direction.

It shows that the non-structural elements have contributed significantly to the lateral stiffness.

**Table 5.** First five mode shapes identified analytically of the brick walled and coated-wall case: (a) 1st Transverse mode, (b) 2nd Longitudinal mode, (c) 3rd Torsional mode, respectively.

| Cases | Measurement Model | Analytical Model |
|-------|-------------------|------------------|
| Brick walled | 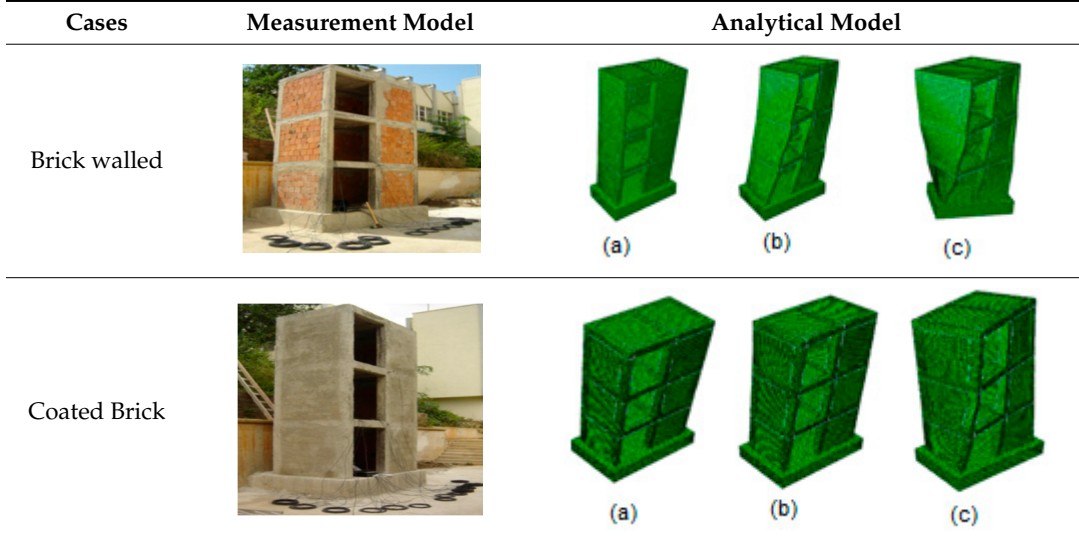 | |
| Coated Brick | | |

### 3.2. Static Analysis

Secondly, the static analyses are conducted under the RC building's self-weight and live loads. A live load of 3.5 kN/m$^2$ is taken into consideration in the analysis as suggested for the design of masonry structure in Turkish load standard TS498 [34]. The displacement and principal stress of the building model varied depending on various construction stages. For a comparison between these values, Table 6 presents the ultimate displacement and principal stress of each construction stage below.

**Table 6.** The ultimate displacement and principal stress values.

| Results | Frame Bare Cases | | | Brick-Wall Case | Coated-Wall Case |
|---------|------------------|------------|--------------|-----------------|------------------|
| | One-Storey | Two-Storey | Three-Storey | | |
| Ultimate displacement (mm) | 1.04 | 1.11 | 1.17 | 1.36 | 1.25 |
| Maximum principal stress (MPa) | 3.61 | 2.40 | 1.24 | 3.36 | 3.41 |

### 3.2.1. Ultimate Displacements

In this section, the ultimate displacement is analyzed for all the cases. In the frame-bare cases, the results for first-storey, second-storey and third-storey cases are 1.04 mm, 1.10 mm and 1.17 mm, respectively. These values are slightly different, quasi-proportional to the storey height. According to the same analysis, the construction of infill walls increases the displacement for the third storey from 1.17 mm to 1.36 mm in Figure 5c,d respectively.

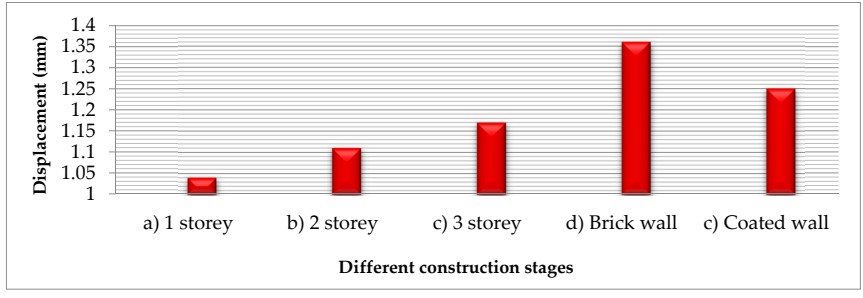

**Figure 5.** Comparison between ultimate displacements at different construction stages.

Under the same static loading conditions, the model with masonry infill walls has significantly greater displacement compared to bare-frame and coated-brick cases. However, the decrease of the displacement value is most pronounced in the coated brick-walled case with 1.25 mm, probably due to the plastering mortar layer applied on the exterior façade of the masonry walls. It can be concluded that the magnitudes of displacements depend not only on the storey's level and masonry walls but also on materials such as plaster coating walls of buildings. The results from the displacement between two models show that the deflections at the top point of the brick walled case are greater than the frame-bare case. It is noticed that the maximum deflection becomes smaller in the middle corners of columns and larger in the connection columns-beams, while in the brick-walled case, the greater deflection is observed at the top left column edge above the infill wall.

### 3.2.2. Maximum Principal Stresses

It can be observed that the maximum principal stress component values obtained from the one-storey RC frame-bare structure were nearly one-and-a-half times and three times, respectively, larger than the two storey and three-storey model buildings. The accumulation of maximum stresses has mostly occurred at the bottom of the corner column and the connection between beams and columns.

The maximum value obtained from the coated brick model is 3.41 MPa, a slight difference compared with that recorded from the uncoated brick case, determined as 3.36 MPa, which is shown in the Figure 6. Even if the stresses that occurred on coated brick walled model are slightly high, it is much less than the crushing limit of the masonry. The values of the average maximum principal stress for masonry units (10 MPa) and minimum principal stress (1 MPa) are reported by Ural and Dogangün [35].

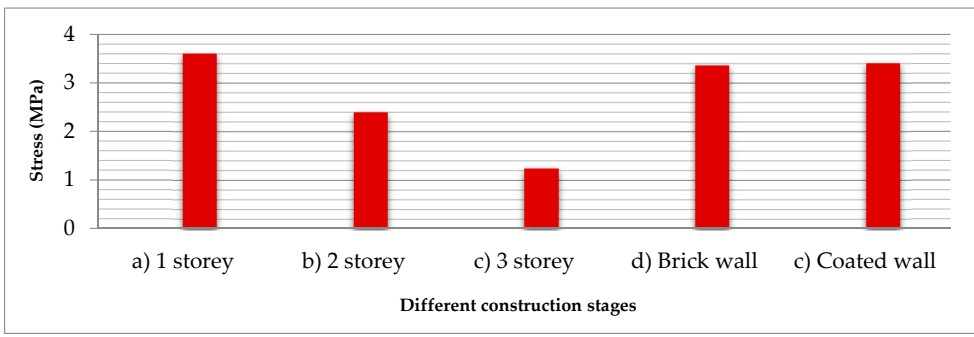

**Figure 6.** Comparison between maximum principal stresses at different construction stages.

### 4. Conclusions

In the present study, the dynamic behaviors of various stages of construction, namely the frame-bare, brick-walled and coated-wall cases are investigated one at a time by using finite element software ABAQUS with the available experimental test data in the literature. In the analysis, the main stages of construction are developed as 3D solid and homogeneous elements. The optimum mesh size was found to be 75mm. The captured conclusions are summarized below:

1.  For the first three modes, the relative errors between the frequencies obtained numerically and experimentally were around 1% to 5%. It indicates that natural frequencies obtained from ABAQUS agree well with the experimental results. However, at the fourth and fifth mode, the difference was between 12% to 16%. This could be ascribed to different reasons; for example, plastering defects or else uncertainties in mechanical properties may be due to the incorrect positioning of samples during laboratory tests.
2.  As a result of this study, it is clearly shown that the presence of infill walls that have plaster work significantly increases the natural frequencies by about 70%, which affects the dynamic behavior of the building, and they must be taken into account in structural analysis.

3. By comparing the displacements for each stage of construction analysis, it can be concluded that the magnitudes of the displacements depend not only on the storey's level and masonry walls, but also on materials such as plaster coating in infield RC buildings. Moreover, it is clearly seen that the maximum principal stress shows no correlation with the masonry unit type. However, the application of a plastering mortar layer seems to increase to some extent the maximum stress.

Further studies should examine the effect of the plaster work on other types of masonry units under seismic performance by taking into consideration the soil parameters.

**Author Contributions:** All authors contributed to every part of the research described in this paper.

**Funding:** The first author acknowledges the support from the Presidency For Turks Abroad And Related Communities under a grant in the scope of Doctoral Fellowship Program.

**Conflicts of Interest:** The authors declare no conflicts of interest.

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
