# Peer review of "Numerical Modeling of Masonry Infilled Reinforced Concrete Building during Construction Stages Using ABAQUS Software"

_buildings, doi:10.3390/buildings9080181_

Round 1
Reviewer 1 Report
This paper attempts to validate operational modal analysis carried out by Türker and Bayraktar (2016) by comparing the result of the aforementioned study with a finite element model developed on Abaqus. However, the authors fail to clearly mention and properly cite the above-mentioned research study in this paper. There is also a high percentage of work that is directly copied from various papers, the similarity index from Turn-it-in with 5 words limit is more than 25% which is not acceptable for a reputable journal.
1. The manuscript is not clear to follow; its aim and objectives are not clear to the readers. It is highly recommended to perform comprehensive proofreading for eliminating grammar and spelling errors.
2. Similarly, capital letters should be used correctly, i.e. for special names, organization, etc.
3. All abbreviations should be mentioned when it appears for the first time.
4. Conclusion Number 3 is vague. Please elaborate.
5. The title of the study is unclear and needs revision.
6. Experimental and analytical mode shapes are compared using Modal Assurance Criterion (MAC). Based on the value of MAC, the authors can conclude whether the mode shapes comparison was accurate. Please refer to the following study for the comparison of operational modal analysis with analytical modal analysis using finite element software.
https://www.mdpi.com/2624-6511/2/1/2
7. The effects of soil surface interaction on the mode shapes are not discussed. The authors can refer to https://www.icevirtuallibrary.com/doi/abs/10.1680/jstbu.18.00131
8. The literature review is weak with very fewer papers from recent years, therefore it is recommended that a thorough literature review be done to improve the quality of the paper.

Author Response
Thank you for your comments, a general correction of the remarks and suggestions that were made in your reviewer report about the article.
The authors have correct and current correctly cite the sources mentioned above in this document and the percentage of articles using the ithenticate program is greater than 20%.
1- The objectives were made clear, and the article was revised by a professor which has comprehensive proofreading in English.
5- The title was revised and modified using keywords with relationship with the subject
6 - Due to reasons beyond our control (the absence of the complete measurement file taken from vibration measurements), the modal assurance criterion (MAC) analysis could not be performed.
7- The model was built in a laboratory and is based on a rocky ground. The soil can be considered as hard. Therefore, in this study, the soil factor is not taken into consideration.
8-New articles from the last years have been added to reinforce the literature section.

Reviewer 2 Report
The authors could consider the following few comments:
General comment: The objective of the study should be clarified. It could be described better what is validating what here: the lab-experiments should validate the FEM, and the FEM is based on something else that is considered to be accurate? I also think the manuscript could benefit from an extra proof-reading by someone with good English skills. It is understandable, but there are some smaller flaws.
Title: Language-wise, should it be something like "Analytical verification of basic experimental procedures used for ambient vibration testing of different construction stages of buildings"? I am not native in English, but I think the title is missing some words.
Introduction: You write: "Furthermore, most of the researches were done only in the laboratory without being validated by Finite element models", should it be "...compared to the Finite element modeling"? Usually are the real tests used as external validation of the models, it is not the models that are validating reality? If it is known that the FEM is correct, then it should be emphasized that the experimental work here aims to see if the test method comply with the FEM (which is "known" to be reliable (?))
Materials/methods: There is no headline that says "2 Materials and methods" or "2 Experimental procedure". I assume that the sub header "The Masonry Building Prototype" is where section 2 starts.
Results/discussion: It is written: "Although the detailed finite element model is developed based on the field survey and engineering judgments ..." This sentence should maybe be incorporated to the end of the introduction (objectives of the study), if the objective of the study is to see how laboratory tests comply with the FEM, which is based on field surveys and engineering judgements.
Author Response
Dear Reviewer 2,
Thank you for your valuable comments and suggestions.
General comment: The objective of the study was clarified and modified so that the readers easily understand the aims of this article. For the validation of the model finite element, in the first time during the writing, I mixed between analytic and the experimental, this fault of the writing was corrected, and at last for errors made in English the article was checked by a teacher with good English skills.
Title: The title was revised and modified using keywords with relationship with the subject
Introduction: We corrected the phrase, the bad writing of the idea was that a contrary meaning was written, Most of this part has been fully rewritten and the title changed accordingly .Yes, you are right the model in finite element must be validated by the experimental and not the opposite, because the experimental guarantee realistic and accurate results.
Materials / methods and Results parts: the errors were corrected and the phrase "Although the detailed finite element model is developed based on the field survey and engineering judgments... '' was rewritten and placed in the introductory part as you suggest, and about the "headline number" in the different headline in the paper has been corrected.
Regards

Round 2
Reviewer 1 Report
The authors have done an effort to review the original manuscript. However, there are still some srious flaws which need to be considered to justify its publication.
The title in revised manuscript still needs to be reviewed, e.g., with "Reinforced Concrete" there needs to be a noun like building,frame, structure etc.
At several places sentences are added like "[3] Concluded that one". This needs to be corrected, either sentence can be modified so that reference comes at the end of sentence, or start the sentence with author-name or words like "Researchers [3]" etc.
"The results from ambient vibration measurements are normally considered to be almost exact." Authors need to justify this sentence, because usually ambient vibration measurements contain a significant amount of noise and in order to get colse-to-exact vibration data extensive pre-processing is needed.
The research significance is still not clear, authors need to elaborate the novelty of the study as a subsection under introduction section.
Author Response
Dear Reviewer 1,
Please see the attachment.
cordially
